# LLM-Based Social Simulations Require a Boundary

## Abstract

This work argues that large language model (LLM)-based social simulations should establish clear boundaries to meaningfully contribute to social science research. While LLMs offer promising capabilities for modeling human-like agents compared to traditional agent-based modeling, they face fundamental limitations that constrain their reliability for social pattern discovery. The core issue lies in LLMs' tendency towards an "average persona" that lacks sufficient behavioral heterogeneity, a critical requirement for simulating complex social dynamics. We examine three key boundary problems: alignment (simulated behaviors matching real-world patterns), consistency (maintaining coherent agent behavior over time), and robustness (reproducibility under varying conditions). We propose heuristic boundaries for determining when LLM-based simulations can reliably advance social science understanding. We believe that these simulations are more valuable when focusing on (1) collective patterns rather than individual trajectories, (2) agent behaviors aligning with real population averages despite limited variance, and (3) proper validation methods available for testing simulation robustness. We provide a practical checklist to guide researchers in determining the appropriate scope and claims for LLM-based social simulations.

## 1 Introduction

Social simulation is a modeling tool that employs computational methods to understand social phenomena. Computational methods, particularly those modeling interactions between individuals, demonstrate advantages in capturing the complex and nonlinear behaviors typically inherent in social phenomena (Eidelson, 1997; Remondino et al., 2010; San Miguel et al., 2012). Among these, Agent-Based Modeling (ABM) is a widely used technique in this area, simulating how individual behaviors and local rules give rise to macro-level patterns (Bonabeau, 2002; Epstein, 1999; Schelling, 1971). ABM offers a bottom-up modeling approach, supports heterogeneity among agents, allows for the exploration of emergent phenomena, and provides researchers with interpretable mechanisms linking micro- and macro-level behaviors (Jackson et al., 2017; Page, 2012; Reeves et al., 2022). Meanwhile, it is controversial due to its reliance on simplification (Edmonds & Moss, 2004), limited adaptability (Wu et al., 2023), sensitive to initial conditions (Manzo & Matthews, 2014), and challenges in representing subjective or human-like behaviors (Ma et al., 2024; Puig et al., 2021), diminishing the contribution of social simulation methods to social science (Reeves et al., 2022).

Recently, LLM agents and social simulations have attracted growing attention. Existing studies have applied LLM agents to domains such as economics (Han et al., 2023; Li et al., 2024), education (Zhang et al., 2024d), game theory (Sreedhar & Chilton, 2024), and social networks (Wang et al., 2023; Yang et al., 2024c; Zhang et al., 2025), with claimed advantages like handling natural language, enabling flexible behaviors, and showing human-like reasoning. However, concerns have also been raised: LLMs may carry social and cognitive biases (Mohammadi, 2024; Navigli et al., 2023), lack behavioral diversity (Ma et al., 2025), and are hard to validate or explain (Larooij & Törnberg, 2025; Ma et al., 2024). Whether or not using LLMs is a good protocol for social simulations remains a question—or may not even be the central question to ask. Many existing studies focus primarily on the simulation itself, while we argue that this narrow focus limits the method's contribution to advancing social science. Before moving forward with more LLM-based social simulations, two critical questions remain:

1. **How can LLM-based simulations benefit studies of social science?**
2. **Can we draw a line to identify what types of problems are suitable for LLMs to solve?**

In this paper, we take the viewpoint that social simulation benefits social science primarily through uncovering social patterns and generating hypotheses. Achieving this requires simulations with high

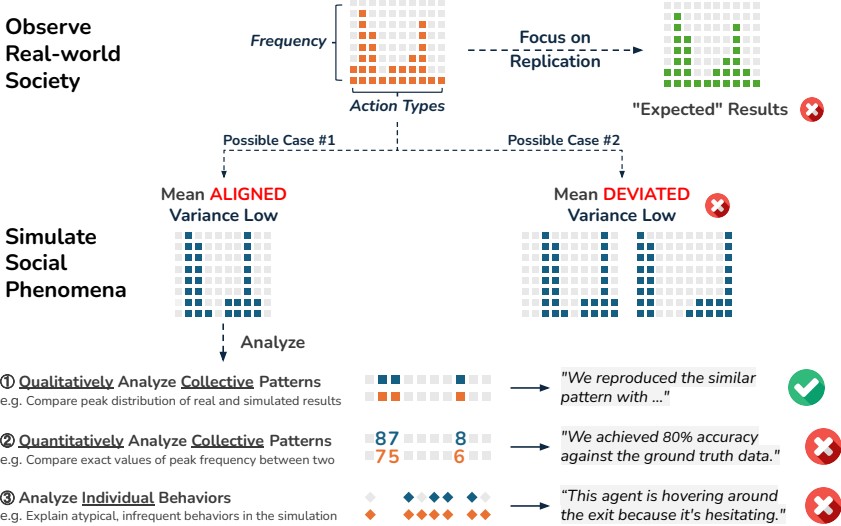

Figure 1: Overview of our claims. We value the goal of social simulations as a means to advance social science, e.g. by explaining social patterns, instead of focusing on "perfect" replication of real-world societies. We further examine possible simulation scenarios (e.g., aligned or misaligned means and variances) and advocate for a stronger emphasis on qualitative analysis of collective patterns.

fidelity and robustness. We emphasize fidelity in particular, examining how alignment, homogeneity, and especially heterogeneity shape social dynamics, and why individual-level heterogeneity is essential for meaningful social simulation. This perspective helps explain why the limited behavioral diversity of current LLM agents constrains their effectiveness in representing complex, multi-agent societies (Ma et al., 2025; Shrestha et al., 2025). We further investigate common issues in LLM-based simulations—such as behavioral variance, social bias, and outcome inconsistency (Larooij & Törnberg, 2025; Mohammadi, 2024) (Figure 1)—and propose to **regulate the applicable boundaries of LLM agent-based social simulation to enhance its contribution to social science, as our central position**. We argue this as a general checklist for evaluating the use of LLMs in social simulation, rather than a how-to guide for conducting such studies.

**Our Contributions.** This work makes three key contributions. (1) We systematically analyze the boundary problems of LLM-based social simulations—the inherent limitations that fundamentally determine their reliability for social pattern discovery, moving beyond implementation issues to examine what current LLM capabilities can and cannot achieve. (2) We discuss simulation fidelity through the concept of agent heterogeneity, indicating why LLMs' tendency to converge towards common patterns fundamentally limits their capacity to simulate complex social dynamics requiring behavioral diversity. (3) We provide heuristic boundaries and a general checklist for when LLM-based simulations can make real contributions to social science research. Rather than asking whether LLMs can replace traditional agent-based models, we reframe the question to focus on precisely defining the scope of problems where LLMs can meaningfully advance social science, which is a perspective essential for responsible deployment and guiding future research towards socially beneficial applications. We expect that the boundaries of LLM-based social simulations, as outlined in this paper, would help bridge the gap between AI and social science sectors and contribute to findings in social science research.

## 2 LLM-BASED SOCIAL SIMULATIONS

### 2.1 OBJECTIVES OF SOCIAL SIMULATIONS

The **primary objective of social simulations** is not to *replicate* reality in fine detail, but to serve as a research tool for scientific endeavors, specifically for explaining social patterns, constructing theories, and providing interpretable foundations for hypothesis generation (Axelrod, 1997; Silverman & Bryden, 2007; Silverman et al., 2018). A clear modeling objective is essential for guiding methodological choices, which can vary significantly depending on the simulation's intended purpose. When objectives are poorly defined, effective validation becomes difficult, particularly when testing alignment with reality and ensuring reproducibility, both of which are critical for establishing credible simulations (Arnold, 2014; Axelrod, 1997; Edmonds & Hales, 2003; Edmonds et al., 2019). To clarify

social simulation's boundaries from the perspective of modeling objective, we examine two modeling objectives frequently declared in LLM-based simulations: replication and prediction. We argue these should not constitute primary goals and may even obstruct effective social science discovery.

Replication-oriented work is common in LLM-based simulation literature, yet studies achieving novel, valuable social science discoveries remain extremely limited. This approach suffers fundamental flaws. Critics note that replication merely repeats known behaviors without revealing new social dynamics or mechanisms (Cheng et al., 2023), contradicting social simulation's core purpose. Schelling's model, a classical model in this field, exemplifies the alternative (Schelling, 1971): through simple, verifiable interaction rules, it demonstrates universal mechanisms of community segregation without replicating any specific community, revealing broadly applicable social patterns. This suggests that *reproducing* real-world social patterns through simple rules requires no precise *replication* of the world to provide explanatory insights and causal understanding. Furthermore, pursuing exact replication increases parameters and artificial assumptions, risking data overfitting, and reducing model verifiability (Larooij & Törnberg, 2025). This creates complex "artificial societies" increasingly detached from reality (Silverman et al., 2018). Additional computational constraints and complexity of sensitivity analysis further obstruct precise replication and reproduction (Borgonovo et al., 2022; Surve et al., 2023). Hence, social simulation neither needs nor can completely replicate reality; focus should center on reproducing and validating key behavioral patterns consistent with real social phenomena (Casti, 1996; Edmonds et al., 2019; Silverman et al., 2018). Only findings from testable reproduction apply meaningfully to social science problems.

Another misconception involves emphasizing *predictive capabilities* through detailed replication performance. Despite numerous attempts at LLM-based social prediction, evidence shows limited performance in predicting social dynamics without oracle information, neither discoveries on effective methods for prediction improvement (Gui & Toubia, 2023; Yang et al., 2024a; Ziems et al., 2024). Critics argue that social simulation predictions often constitute mere retrodictions of existing patterns, lacking effective future scenario generalization (Edmonds, 2023; Polhill et al., 2021). Predicitive capabilities are further undermined by the flaws and biases in LLM-based simulations. For example, using retrodictive tests to claim predictive capabilities (Wang et al., 2025d) may introduce data leakage, as the retrospective scenarios could already be contained within the LLM's training data; such bias is hard to eliminate because the LLM could infer the scenario, despite the removal of time, location, and persons involved in the incident from the prompt, and implicitly use its knowledge to make "predictions". Many simulation works' predictive claims thus exceed actual model capabilities, which requires enhanced validation (Ball et al., 2024; Cao et al., 2025; Chuang et al., 2024b; Orlikowski et al., 2025; von der Heyde et al., 2024; Wang et al., 2025b; Yang et al., 2024a; Zhang et al., 2025). Nevertheless, few studies establish reliable validation methods and sensitivity analyses (Chatterjee et al., 2024). Moreover, current works claim that simulations reflect real social dynamics (Yang et al., 2024c; Zhang et al., 2025), based on simple validation efforts such as LLMs' explanation of their own decision-making process, which might raise endogeneity issues. Whereas it is crucial to create comprehensive frameworks for simulating social phenomena using LLM agents at unprecedented scales, as achieved in these works, researchers need to be cautious with their objectives and findings, because misguided objectives and overstated findings will prevent social simulation from fulfilling its promise in social science research.

In sum, social simulation's limitations stem from both LLMs' inherent capabilities and simulation framework design issues (Wang et al., 2025c). We thus emphasize caution regarding simulation modeling with replication and prediction as core objectives, advocating instead for greater focus on simulation alignment with key social patterns and its validation.

## 2.2 Current Challenges that LLM-Based Simulations Face

Now that we have distinguished the fundamental purpose of social simulation from replication and prediction purpose, we turn our attention to the specific issues confronting LLM-based social simulations. We categorize these challenges into two primary areas: (1) **usage problems**, which pertain to how researchers apply LLMs in simulations and whether these applications align with effective simulation practices; and (2) **boundary problems**, which relate to the inherent subjective capabilities and limitations of LLMs themselves, discussing what LLM-based simulations can and cannot reliably achieve. This paper will primarily focus on the latter, the boundary problems, as they fundamentally determine the reliability and applicability of LLM-driven social pattern discovery.

**Usage Problems: Misuse of LLM-Based Simulations**    Usage problems arise from the way LLMs are employed in social simulation designs. A common issue, as mentioned in Section 2.1, is the tendency for simulations to aim for perfect replication of reality. Such an objective is not only inherently

difficult but can also undermine the capacity for meaningful social pattern discovery (Edmonds, 2023; Hassan et al., 2013). Overly precise replication sometimes introduces researchers' subjective judgments or requires extensive data to calibrate simulations at the micro level (Bertoni, 2023; Paudel & Ligmann-Zielinska, 2023; Yarkoni & Westfall, 2017).

Beyond the fundamental purpose, other common usage problems specific to LLM-based simulations include, but are not limited to: (1) imprecise or self-evident prompt engineering that can lead to simulation distortion (Mannekote et al., 2025; Ronanki et al., 2024); (2) overly large or ill-defined action spaces for LLM agents, which often result in the generation of invalid behaviors, complicate rigorous sensitivity analysis, and amplify errors across multiple iterations (Guo et al., 2024; Liu et al., 2024b;c; Yim et al., 2024); and (3) simulation frameworks that introduce excessive researcher assumptions or constraints, inadvertently causing models to lose their emergent capabilities (Silverman & Bryden, 2007). While these usage issues significantly impact the effectiveness of social simulation, this paper will **NOT** primarily focus on problems caused by researchers' subjective choices or design flaws that could, in principle, be mitigated by better practices. Our scope specifically targets the intrinsic limitations of LLM technology itself.

**Boundary Problems: Inherent Limitations of LLM Agents**    Boundary problems constitute the core focus of this paper, as these boundaries fundamentally determine the reliability of social pattern discovery derived from LLM-based simulations. These problems represent the inherent, subjective limitations of current LLM technology when applied to social simulation. Clarifying these boundaries is essential for understanding where LLM-based social simulations can reliably contribute. We specifically examine three critical aspects that collectively define the scope of LLM-based social simulations:

1. **Alignment (Sections 3 and 4):** This concerns whether the simulated agents' behaviors and collective dynamics are aligned with real societal patterns. This aspect primarily affects whether such simulations can genuinely be used to understand real social phenomena, as discussed previously. Alignment is our main focus in this paper, where we delve deep into the types of alignment, what is currently lacking in LLM agents, and what we can reliably claim and simulate.
2. **Consistency (Section 5):** This refers to whether the simulated agents can maintain fidelity to their assigned roles and behavioral patterns over a long temporal horizon. Social simulations often span extended periods, but LLMs inherently face challenges with long-context understanding and coherent behavior over time. Ensuring a consistent simulation throughout an entire episode is crucial for deriving reliable conclusions.
3. **Robustness (Section 6):** This addresses whether the simulated society is reproducible and stable under different prompt settings, initial conditions, or minor perturbations. Robustness directly impacts the reliability of the simulation's findings, which is paramount for any subsequent analysis and valid pattern discovery.

We will proceed by discussing these three aspects of LLM-based social simulations in the aforementioned order, prioritizing the intricate challenges related to alignment. By analyzing these three aspects, we aim to precisely delineate the boundaries of the scope of social problems and the validity of related claims that can be researched under current LLMs' capabilities.

## 3    ALIGNMENT AND HETEROGENEITY

The degree of alignment between LLM-based simulations and real-world behavior is a key factor in determining the reliability of insights drawn from social pattern discovery. This alignment can be further divided into two aspects: **individual-level alignment**, which concerns whether each agent behaves in a human-like manner, and **collective-level alignment**, which concerns whether the interactions among agents reproduce realistic social dynamics and emergent phenomena. These two aspects are interrelated, and understanding their respective roles is essential before applying LLMs to simulating social phenomena.

**Relative Importance of Individual Alignment**    While individual-level alignment is often desirable, perfectly capturing individual behavioral patterns is not always essential for obtaining conclusions with practical utility in social simulations. This is because social phenomena emerge primarily from interactions between individuals rather than from individual behaviors alone. As Durkheim (2023) argued in his foundational work on social facts, collective phenomena possess properties that cannot be reduced to individual psychological states. Building on this insight, while reductionist approaches focus on individual-level fidelity, the emergent properties of social systems cannot be fully predicted from the knowledge of individual components alone (Holland, 2000; Louth, 2011; Squazzoni et al.,

2014). Studies in computational social science have demonstrated that weak individual alignment can still lead to the emergence of complex collective behaviors: Granovetter (1978)'s threshold models show how individual decisions with simple thresholds can produce unpredictable collective outcomes, while Reynolds (1987)' boids model demonstrates how complex flocking behaviors emerge from just three simple rules governing individual agents' separation, alignment, and cohesion. These findings suggest that individual-level fidelity is neither the sole nor the primary factor in generating realistic social dynamics. On the other hand, approximate individual-level modeling can still capture the essential social interaction dynamics. For instance, an LLM-based simulation that reproduced the aforementioned Schelling's model demonstrated that highly segregated societies still emerge even when LLMs exhibit relatively low bias, with simple behavior settings, decision methods, and a degree of individual *homogeneity* (Cheng et al., 2024). In this setup, the final social structure is largely independent of specific individual intentions or detailed behavioral trajectories. This illustrates that the emergence of collective patterns can be relatively insensitive to individual-level modeling imperfections, suggesting that strict individual alignment, while beneficial, is not uniformly the most critical factor for valid social simulations focused on macroscopic phenomena.

**Homogeneity and Collective Alignment**    To further explore collective alignment, it is necessary to understand the interplay between individual *homogeneity* and *heterogeneity* within a system, as these properties of agents become apparent through their interactions. *Homogeneity*, characterized by agents sharing similar traits or behaviors, can, in certain cases, lead to emergent social patterns. As previously noted in the Schelling's model example, even simple, uniform preferences can result in collective phenomena such as segregation.

However, when agents are highly homogeneous in their decision-making rules and responses, the resulting collective behaviors tend to converge to predictable equilibrium states that can be analytically characterized. For example, in voter models where all agents follow identical imitation rules, the system predictably converges to consensus with mathematically derivable convergence rates and final outcome probabilities (Castellano et al., 2009; Holley & Liggett, 1975). Similarly, in simple social contagion models where individuals adopt behaviors through independent exposures with uniform transmission probabilities, the spread patterns follow predictable epidemic trajectories characterized by standard parameters such as peak timing and final adoption rates (Hodas & Lerman, 2014; Sprague & House, 2017). The scope and complexity of patterns that can emerge solely from homogeneous interactions are often limited.

Due to this limited complexity arising from homogeneous interactions, collective behaviors driven primarily by homogeneous agents can often be adequately characterized through aggregate statistical analyses, obviating the need for complex bottom-up interactive simulations (Galla et al., 2006; Galstyan et al., 2005; Helfmann et al., 2021).

**Critical Role of Heterogeneity**    Conversely, *heterogeneity* is widely recognized as a fundamental driver of complex social dynamics and intricate emergent phenomena. Existing work across various contexts has consistently reported that certain emergent phenomena only occur with sufficient diversity among agents, a domain where traditional rule-based simulation methods like ABM often face limitations (Deter & Sayama, 2024; Gao et al., 2024). The importance of agent *heterogeneity* has been emphasized in numerous studies, spanning computational simulation directions (e.g., social network modeling (Ojer et al., 2025), epidemic intervention (Lorig et al., 2021; Reeves et al., 2022), climate change policy (Mercure et al., 2016), and wealth formation (Wang et al., 2010)) and problem-solving applications (e.g., multi-agent cooperation analogous to human dynamic collaboration (Chen et al., 2024) and multi-agent software development (Hong et al., 2024; Qian et al., 2024)).

**Heterogeneity v.s. Homogeneity**    From a complex systems perspective, collective behavior fundamentally differs from simple aggregations of individual behavior. When individual differences exist, interactions create feedback mechanisms that may amplify these differences, producing emergent collective phenomena that cannot be predicted from average individual characteristics (Miller & Page, 2009). While heterogeneity enables rich individual interactions that generate intricate patterns and structural biases (Amin et al., 2018), homogeneity tends to average out these behavior, limiting emergent complexity (Maciejewski et al., 2014).

Consider two illustrative cases. In social choice theory, the Condorcet Paradox demonstrates how diverse individual preferences can produce collective voting cycles—collective behavior outcomes that cannot be understood by simply averaging individual preferences (Gehrlein, 1983). From another side,

when we assume perfect homogeneity of agents in economic models (i.e. identical rationality and complete knowledge in "Homo economicus"), the ideal simulated market will reach immediate equilibrium with zero average profits, precluding the market dynamics that define real economic systems (Grossman & Stiglitz, 1980). These examples show that while homogeneity can yield certain patterns, it fundamentally limits the emergence of rich, complex dynamics characteristic of real-world social systems.

**Implications for LLM-Based Simulations**    In summary, these considerations illustrate that neither perfect individual alignment nor homogeneous interactions alone are sufficient for capturing complex social dynamics. The ability of social simulations to discover and accurately reflect novel, complex social patterns largely depends on the degree of *heterogeneity* among agents. Consequently, **whether the behavior of collectives composed of LLM agents can reflect "sufficient" heterogeneity** becomes a critical indicator of simulation validity. If the phenomena under investigation fundamentally require sufficient heterogeneity for their emergence, yet LLMs inherently represent insufficient diversity among individuals, then the conclusions drawn from such simulations may not reliably apply to real-world situations. The subsequent discussion will systematically examine how heterogeneity may be lacking in existing works on LLM-based simulations and the potential distortions this absence may introduce.

## 4 LLM-BASED SIMULATIONS LACK HETEROGENEITY

### 4.1 THE "AVERAGE PERSONA": ORIGIN AND DIMENSIONS OF LIMITED HETEROGENEITY

As established in the previous discussion, the capacity of agents to exhibit sufficient heterogeneity is important for social simulations aiming to reveal novel and complex social dynamics. Current LLM agents basically fall short in generating such necessary diversity. This problem is often reflected in their tendency to act as an "average persona". This average persona reflects LLMs' built-in bias to converge towards common patterns. The argument advanced in this paper is that the impact of this average persona on heterogeneity can be analyzed through two key behavioral dimensions: **variance** (representing the diversity and spread of behaviors) and the **mean** (representing the central tendency or average behavior, and its alignment with human behaviors). We propose this variance-mean decomposition as a useful framework for diagnosing different types of alignment problems, where variance captures whether LLMs can generate the behavioral diversity necessary for complex social dynamics, and mean alignment determines whether the central tendency of LLM behavior corresponds to real human populations. This analytical approach enables us to categorize the specific limitations of LLM-based simulations and establish appropriate boundaries for their application.

This "average persona" phenomenon can be understood through the lens of the models' training processes. A key contributing factor appears to be that language model training maximizes the conditional probability of predicting text through likelihood-driven loss functions over vast human expression data. This training objective inherently rewards high-frequency, mainstream expressions and suppresses marginal ones, thereby fostering an "average persona" that aggregates group thinking and limits distributional representativeness (Dung Nguyen et al., 2025; Trott, 2024; Wang et al., 2025a). The inherent heterogeneity of subgroups is consequently erased, causing behavior to concentrate on a few dominant patterns that often reflect social biases and demographic stereotypes, even when instructions attempt to elicit alternative perspectives (Liu et al., 2024a; Taubenfeld et al., 2024). This results in the difficulty in capturing long-tail patterns, even with advanced simulation frameworks or model improvements (Taubenfeld et al., 2024; Wang et al., 2025a). We delineate two primary cases based on how this average persona manifests through **variance** and **mean**, each with distinct consequences for social simulations.

### 4.2 APPLICABILITY AND CLAIM BOUNDARIES IN LLM-BASED SOCIAL SIMULATIONS

**Case 1: Low Behavioral Variance, Mean Aligned**    In the first case, LLM agents exhibit a low behavioral variance, meaning their strategies and actions are concentrated, lacking the broad diversity observed in human populations. However, the mean (average behavior) of these agents may still align reasonably well with the average behavior observed in real-world human experiments.

Existing work consistently notes that LLMs generate insufficient diversity and exhibit overly homogeneous behavior, often missing human randomness and error patterns (Aher et al., 2023; Anthis et al., 2025; Cheng et al., 2023; Lau et al., 2024). For instance, in economic market simulations, while LLM agents can replicate macroscopic patterns observed in human experiments, individual LLM agents demonstrate significantly less behavioral variance, employing more concentrated strategies compared to diverse human participants (del Rio-Chanona et al., 2025; Han et al., 2023). Similarly, in

the Keynesian Beauty Contest (KBC, guessing 2/3 of the average), LLM-based simulations successfully reproduced several peak guess values consistent with human experiments, but the frequency of guesses on non-high-frequency values was markedly lower than that of real human participants (Figure 2) (Wu et al., 2024). Even in emergency evacuation simulations, despite group-level differences based on personas, individual agent trajectories could be surprisingly similar (Wu et al., 2023). These examples illustrate that models may converge towards unified, high-frequency answers that align with human values but diverge from the full behavioral distribution of real humans (Aher et al., 2023).

The phenomenon here is that even with limited intra-group behavioral differences among LLM agents, if their collective behavioral patterns are meaningful and consistent with real-world aggregated outcomes, insufficient variance does not always undermine the purpose of the simulation at the macroscopic level. However, this scenario mandates strict examination of the boundaries of claimed findings. **Researchers should focus on the collective behavior and qualitative patterns, as these may be well-reflected despite low individual variance.** Conversely, attempting to interpret the significance of individual agent "behavioral trajectories"—such as specific decisions in an economic market or particular paths during an evacuation— can lead to "interpretive overfitting". This is because individual decisions may not align with real-world situations (e.g., for personal mobility simulation, real-world activities are observed to be less frequent than simulated ones in the COVID-19 pandemic scenario (Wang et al., 2024a)), and it is difficult to verify their underlying reasoning or distinguish them from potential hallucination (Singh et al., 2024). Thus, while exploring specific agent decision logic might enhance understanding from an AI/ML perspective (e.g., k-level reasoning in KBC (Gandhi et al., 2023; Zhang et al., 2024b)), its significance for social science is relatively low when individual variance is constrained.

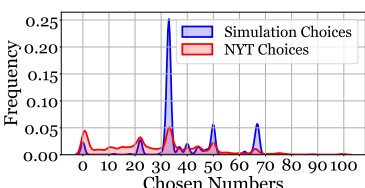

Figure 2: Distribution of chosen numbers by GPT-4 (in blue) vs. by humans (in red), adapted from KBC (Wu et al., 2024). The LLM successfully reproduces several peak guess values (e.g., 33, 50, and 66), closely aligning with human choices. This indicates that the LLM has aligned well with the human **mean**. Meanwhile, the frequency distribution for less common values shows inconsistencies compared to human behavior, highlighting the LLM's low behavioral **variance** in the simulation.

**Case 2: Low Behavioral Variance, Mean Deviated**   The second and more critical case arises when LLM agents exhibit not only low behavioral variance but also a mean (average behavior) that deviates significantly from human values or real-world distributions. This deviation means the aggregated behavior of LLM agents does not accurately reflect the central tendency or typical actions of the human population or subgroup they are intended to represent.

The consequences of this mean deviation are profound for social simulations. Unlike Case 1, where some insights into average collective patterns might still be gleaned, this scenario can render LLM-based social simulations problematic or even inapplicable for deriving meaningful insights into real human societies. For example, research has found that LLMs perform significantly differently when simulating various population subgroups, often exhibiting biases not present in the intended human population (Ma et al., 2025). In public opinion surveys, models trained with human feedback tend towards liberal views and exhibit more polarized attitudes, proving difficult to debias through role-play (Bernardelle et al., 2024; Bisbee et al., 2024; Santurkar et al., 2023). Such deviations are widespread across models and contexts; generated dialogues often differ from real human conversations in linguistic features and exhibit lower diversity (Lin et al., 2024). Moreover, training processes that aim to debias or rationalize certain LLM behaviors, while highly valuable for general applications, can paradoxically compromise the simulation's utility for studying certain social phenomena. When the research objective specifically requires understanding how biases and irrational behaviors contribute to social patterns, their elimination becomes problematic rather than beneficial. For instance, humans exhibit response biases to specific survey wording, whereas models can be less sensitive to such perturbations, failing to capture behavioral mechanisms that may be central to the phenomena under investigation (Tjuatja et al., 2024). Deviations are also evident in cultural contexts; multilingual simulations of the trolley problem have shown LLM agents making moral judgments inconsistent with the cultural values of the human communities speaking those languages (Jin et al., 2024). This inability to adhere to nuanced linguistic and cultural conventions is a widespread phenomenon (Naous & Xu, 2025; Zhang et al., 2024c).

Therefore, when such mean deviation exists, the utility of LLM-based social simulations for social science insights becomes severely limited. **Researchers must diligently check for the presence of such deviations.** If the average behavior of LLM agents demonstrably diverges from the actual

average behavior of real human populations in the studied context, then the simulation's applicability for reflecting real human society is significantly compromised or even negated. Achieving greater alignment often requires constructing extensive detailed socio-demographic conditions to personalize LLMs (Argyle et al., 2023), yet the reasons for significant deviations from human preferences can remain unknown (Dung Nguyen et al., 2025).

**Challenges in Enhancing LLM Agent Heterogeneity**  Various existing methods aim to construct personalized, diverse, and reality-aligned agents, including prompt engineering (Park et al., 2022), personality measurement-based prompting and alignment (Serapio-García et al., 2023), character modeling architectures that generate profile copies of real people through interviews or questionnaires (Jung et al., 2025; Park et al., 2024), and alignment based on large-scale data (Ge et al., 2024; Li et al., 2025). However, these approaches face significant limitations. Prompt engineering often cannot completely eliminate bias (especially for minority groups, a critical social science concern), while alignment dependent on large-scale datasets is costly, and high-quality personalized preference data remains scarce, with anonymity issues affecting generalization capabilities (Li et al., 2025). As the number of individuals in a simulation increases, the cost of such detailed modeling rises dramatically, often forcing trade-offs between individual modeling precision and simulation scale. This can inadvertently limit simulation boundaries and sacrifice heterogeneity for controllability in large-scale scenarios (Chen et al., 2025; Mou et al., 2024). Some works have also used various LLMs to attempt to enhance heterogeneity, while acknowledging that agent behaviors still concentrate on a few strategies, reflecting limited heterogeneity (Fontana et al., 2024; Lu, 2024).

Furthermore, achieving alignment at the individual level does not necessarily guarantee that collective behavior will also align with the real situation, as previously distinguished. Bias removal methods, while enhancing fairness, may simultaneously weaken knowledge maintenance and overall model performance (Chen et al., 2025). In social simulations, standardized methods to confirm which approaches can truly achieve diverse and aligned heterogeneity construction are still lacking, as is a clear determination of how LLM parameters (e.g., model temperature) should be set for optimal diversity. We cannot guarantee the alignment of all behaviors solely through observations of agent alignment with reality in certain specific behaviors either. The effect of adding personas can even be randomized in some contexts (Zheng et al., 2024). Ultimately, prompting may only capture superficial, stereotypical personas, struggling to penetrate individuals' deep beliefs, values, learning histories, and nuanced decision-making processes. Therefore, researchers must be extremely cautious about the scope of conclusions drawn from LLM-based social simulations, rigorously verifying both the diversity (variance) and alignment (mean) of agent heterogeneity, and determining whether the observed lack of heterogeneity represents merely insufficient diversity or, more critically, a significant deviation from average real-world behavior.

## 5 Consistency in Long-Term Simulations

At the individual alignment level, we need to consider alignment issues in multi-round social simulations. Unlike single-round Q&A, in long-term social simulations, LLM agents, constrained by model capabilities, may fail to maintain cognitive consistency in their roles during extended interactions (Huang et al., 2024). Since LLMs lack the ability for continuous exploration in environments and possess no memory capabilities, they can only respond passively to context sequentially (typically, each API call produces independent responses related only to the current model input, despite pass actions reprompted to simulate the "memory" of an LLM agent). Slight differences and perturbations in context across different rounds may cause the same LLM agent to produce inconsistent reactions (Yao et al., 2023; Zhu et al., 2023). When an agent's behavioral traits are important to the problem (especially from a heterogeneity perspective, where one agent's behavioral traits significantly alter other agents' behaviors), these inconsistency-induced trait changes may significantly alter the macro patterns demonstrated in the simulation. Without careful verification of consistency across the temporal dimension, researchers might misinterpret significant pattern changes as some emergent phenomenon rather than recognizing them as stemming from insufficient capabilities of the LLM.

## 6 Robustness in LLM-based Social Simulation

Robustness refers to whether simulation conclusions can remain stable and reproducible under different parameter settings, uncertainties in simulation conditions, and perturbations. The difficulty of LLMs in providing repeatable results is a major challenge, yet necessary sensitivity analysis practices are rarely implemented in existing studies (Larooij & Törnberg, 2025). In the context of LLM-based simulations,

this varies across problems and modeling approaches, primarily verified through sensitivity analysis to examine whether the qualitative patterns of simulation findings are sensitive to minor differences in context or prompts, and whether changes in context significantly affect agent behavior (Hosseini & Horbach, 2023; Yang et al., 2024b; Ziems et al., 2024). For instance, an LLM's sensitivity can vary significantly; in some situations, LLMs may display excessive sensitivity towards groups or topics that could cause fairness issues, resulting in the misclassification of benign statements as hate speech, while in other contexts, they may achieve a better balance (Zhang et al., 2024a). Whether LLM-based simulations can maintain discovered patterns under perturbations constitutes one boundary of the simulatable range. Currently, this can only be tested through actual sensitivity analysis to verify whether conclusions are reproducible, while lacking a priori methods to explain what the hard boundaries of LLM-simulatable problems are.

## 7 DISCUSSIONS

We have introduced the impact of LLM individual and collective alignment issues, long-term individual consistency problems, and robustness issues on the reliability of social simulation conclusions. These discussions indicate that although LLM alignment with reality is limited in some domains and may contain biases that are difficult to identify and interpret (Shin et al., 2024), LLM-based social simulations can still provide important insights for social pattern discovery and hypothesis generation. The prerequisite is that researchers need to clearly define the **scope of claims that can be declared** from simulation results, as well as the **scope of problems that can be simulated**. We summarize these discussions as *heuristic* boundaries for LLM-based simulations:

**Boundary 1: Collective v.s. Individual Behavior**    The first boundary concerns the level of analysis. LLM-based simulations are more reliable when: (1) Focusing on **collective** patterns rather than individual behavioral trajectories. (2) Studying emergent phenomena that depend more on **interaction** modes than on precise individual characteristics.

**Boundary 2: Alignment and Heterogeneity**    The second boundary relates to the "average persona" phenomenon in LLM agents and its implications for simulation validity: (1) LLMs often manifest an "average persona" due to training processes that favor mainstream patterns, leading to reduced behavioral **variance** and potentially erasing subgroup characteristics or reflecting social biases. Crucially, the simulation's behavioral **mean** might also deviate from actual human population averages which may negatively impact alignment with real societal patterns. (2) When LLM agents show low variance but their mean of collective behavior aligns with real-world outcomes, simulations can offer valuable collective insights. However, when agents exhibit both low variance and mean behaviors, which significantly deviate from the relevant human population, the simulation's findings become inapplicable to real human societies.

**Boundary 3: Temporal Consistency**    For multi-round or long-term simulations, we encourage researchers to consider: (1) Whether agents can maintain consistent patterns that authentically reflect their personas over a **long-term** simulation. (2) Whether observed pattern changes reflect genuine emergent phenomena or artifacts of LLM limitations. In addition, to achieve valid social simulations, validating the robustness of simulation results through sensitivity analysis to exclude simulations where behaviors change significantly under context disturbance, and avoiding erroneous simulation purposes and usage will jointly constitute the boundaries of LLM-based social simulations.

## 8 CONCLUSION

This paper argues that the primary goal of LLM-based social simulations is to explain social patterns, construct theories, and generate hypotheses. Misunderstandings about these goals in current research have limited their contributions to social science. To better address social science problems, we highlight the need to focus on collective alignment and enhance agent heterogeneity to more accurately reflect real societies. Additionally, individual temporal consistency and simulation robustness are equally essential for applying insights from simulated societies to real-world contexts.

Our core standpoint is to **emphasize the necessity of regulating simulation boundaries, including the scope of claims and simulated problems**. We urge the community to treat these boundaries as **a general checklist for evaluating the use of LLMs in social simulations**, thereby ensuring their **positive contributions to social science research**. Meanwhile, we emphasize advancing the standardization of systematic validation methods for social simulations, as well as enhancing the capability to identify potential biases in simulations, to avoid neglect or bias towards marginalized groups and phenomena.

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

## A RELATED WORKS

### A.1 COMPUTATIONAL SOCIAL SCIENCE

Social phenomena typically arise from the interactions of intelligent, adaptive agents under dynamic conditions Eidelson (1997); San Miguel et al. (2012). Even when we fully understand behavior at a small scale (e.g., personal behavior), we may not necessarily understand social phenomena at the macro scale Squazzoni et al. (2014). This complexity presents enormous challenges for social science research, including interpreting causal relationships, determining the applicable scope of problems, and ensuring reproducibility of conclusions. This aligns with sociologist Giddens' proposition that social structures and social practices are interrelated and difficult to find cause-and-effect relationships Giddens (1986a;b); Wheeler-Brooks (2009). Therefore, traditional social science methods—such as surveys and laboratory experiments—struggle to capture the nonlinear and emergent dynamics of real-world social systems, are prone to deriving erroneous patterns from data (known as "apophenia"), and may overlook failure modes not incorporated into the patterns Abell & Reyniers (2000); Bragues (2011); Mondani & Swedberg (2022). These challenges have driven the rise of computational social science, which attempts to use algorithmic, data-driven, and simulation-based approaches to model and interpret complex social behaviors at scale.

### A.2 AGENT-BASED MODELING IN COMPUTATIONAL SOCIAL SCIENCE

ABM has been a foundational method in computational social science, enabling researchers to simulate macro-level outcomes from simple micro-level behavioral rules (Bonabeau, 2002). Classic examples include Sugarscape (Epstein, 1999) and Schelling's segregation model (Schelling, 1971), which illustrate how wealth gaps or segregation patterns can emerge from individual interactions. Despite disagreements and inconsistencies within social science theories, many works agree that social interaction is the fundamental unit of sociological analysis and plays a crucial role in research, rather than focusing solely on individual behavior or macro structures Gerring (2001); Mondani & Swedberg (2022); Turner (1988). By ABM, the modeling of social interaction can fill the gap in this micro-macro linkage.

ABM provides explanatory power through controlled simulations, but its limitations are widely acknowledged. These include reliance on hard-coded rules or heuristics, difficulty in encoding subjective behaviors, poor agent adaptability, and simplification of heterogeneity (Edmonds & Moss, 2004; Reeves et al., 2022; Wu et al., 2023). Moreover, the need for handcrafted agent behavior risks introducing researcher bias, and limits the scalability and generalizability of such models to real-world complexity (Williams et al., 2022).

### A.3 LLMS IN SOCIAL SIMULATIONS

Recently, the emergence of LLMs has reignited interest in agent-based simulation by enabling more natural, flexible, and human-like behavioral modeling. LLM agents demonstrate powerful capabilities in understanding ambiguous instructions, simulating subjective decision-making, and generating explanations in natural language (Adornetto et al., 2025; Ma et al., 2024; Park et al., 2023). They show potential across various social science domains: (1) From the technical perspective, LLMs' powerful natural language capabilities and theory of mind (ToM) capabilities expand the boundaries of traditional simulations. For example, the use of LLM agents enables subjective behavioral modeling and the ability to understand ambiguous natural language instructions Wang et al. (2024b), allows simulation of theory of mind capabilities Ma et al. (2023), enhances interpretability through generative explanations Epstein (2023); Ma et al. (2024), and offers ethical and cost advantages compared to human subject experiments Mou et al. (2024). (2) From the modeling perspective, LLM agents' generalization capabilities can be leveraged to test various scenarios, creating value across interdisciplinary fields Mou et al. (2024) and improving the fidelity of complex behaviors such as interaction, collaboration, and gaming Ma et al. (2024). (3) Exploratory studies have demonstrated human-like behavior, with performance approaching that of humans in certain experiments Anthis et al. (2025).

However, recent criticisms have highlighted significant limitations. LLM agents may inherit and amplify social biases present in their training data Ashery et al. (2025); Mohammadi (2024); Navigli et al. (2023), lack sufficient behavioral heterogeneity Ma et al. (2025), lack human characteristics such as the ability to learn independently and memory Ma et al. (2024), and lack transparency and interpretability due to their black-box nature Larooij & Törnberg (2025). Furthermore, they tend to collapse to high-probability responses, which limits their ability to simulate the diversity of real human behavior, particularly in contexts with high subjectivity or cultural variability Shrestha et al. (2025).

Validating simulation results and their generalizability to real-world phenomena remains a major open question Chuang et al. (2024a); Hua et al. (2023); Lorè & Heydari (2024); Warnakulasuriya et al. (2025). These situations pose challenges in translating the potentials discovered in existing works into findings.

## B    CHALLENGES AND FUTURE DIRECTIONS

The boundaries of LLM-based simulations present several challenges and areas for improvement. **(1) Validation.** While validation of LLM individual behavior and dynamic interactions is more difficult compared to traditional ABM methods, there is currently a lack of good evaluation methods, with heavy reliance on manual or LLMs' self-report approaches for validation (Adornetto et al., 2025; Mou et al., 2024). In response, the simulation community needs to promote systematic evaluation standards to examine whether LLM-based simulations can yield conclusions beneficial for understanding real society. **(2) Conditions of claims.** Social simulation research needs to more rigorously consider the proper claims of simulation conclusions, including clearly defining the conditions under which conclusions hold, their scope of applicability, and their generalization ability in real-world contexts, avoiding overclaims that reduce the credibility and applicability of simulation conclusions. For instance, while simulations with constrained heterogeneity can produce findings consistent with general patterns—such as case studies showing that organizational diversity typically does not improve collective performance—researchers must meticulously bound their claims, as these simulations may fail to capture specific conditions (e.g., extreme individual bias) where the opposite effect occurs (Xu et al., 2014), and dramatically increased heterogeneity may reveal emergent phenomena beyond the original scope. **(3) Bias and ethical concerns.** Close attention needs to be paid to bias issues in LLM-based simulations. Limited by the lack of heterogeneity in LLMs, simulations may lead to neglecting marginalized groups or generating stereotypes and negative biases towards specific populations or phenomena. It is necessary to confirm whether LLMs capture biased "averages" and conduct moral and ethical considerations. **(4) Empirical research.** Considering that our ultimate goal is to contribute to the society, applying findings from social simulations to empirical solutions to real-world problems to confirm or refute the reliability of conclusions may be the next step the community needs to actively take to enhance the credibility and importance of simulation methods in research (Popper, 2005; Watts, 2017).

## STATEMENTS ON THE USE OF LARGE LANGUAGE MODELS

We used LLMs for grammar checking and polishing the English. The authors are responsible for the entire content, which originates from the authors themselves.

