# OpenReview forum: "LLM-Based Social Simulations Require a Boundary"
_ICLR.cc/2026/Conference — Submitted to ICLR 2026_

### Official Review · Reviewer_WPiB · 2025-10-18

**Soundness:** 2
**Presentation:** 3
**Contribution:** 1
**Rating:** 2
**Confidence:** 5

**Summary:**

This paper presents counterarguments against existing LLM-based social simulation work through extensive research and analysis of current approaches. This paper focuses on three key issues: alignment, consistency, and robustness. Building on this foundation, the authors propose what they consider to be the criteria for high-value social simulation, centered around the three aforementioned evaluation dimensions.

**Strengths:**

1. This paper focuses on the rapidly evolving field of LLM-based social simulation and conducts extensive literature research to form its conclusions. The findings provide valuable reference and guidance for relevant researchers.
2. The narrative flow and textual logic throughout the text are coherent and logical, making it easy to read and comprehend.

**Weaknesses:**

1. This paper is purely perspective-based and does not introduce any new techniques or present significant new experiments. It appears to fall outside the scope of the ICLR community and would be better suited for publication in journals or workshops that are more receptive to such work.
2. It is questionable whether the three evaluation aspects of alignment, consistency, and robustness proposed in this paper possess any particular characteristics specific to LLM-based social simulation. First, as emphasized by the author throughout the article, consistency and robustness are fundamental requirements for any simulation system. And alignment with the real world remains the ultimate goal of any predictive or simulation effort. Therefore, how does this paper demonstrate the innovation of the proposed evaluation system, and is this evaluation system unique to LLM-based social simulations?
3. Throughout this paper, certain sections cite literature from before the advent of LLMs as supporting material, which appears to fall outside the scope of the discussion presented herein.

**Questions:**

1. Can an evaluation method be designed based on the proposed evaluation framework to evalulate existing LLM-based social simulation work?
2. Does the evaluation framework proposed in this paper account for the changes introduced by LLM to social simulation?
3. Based on my understanding of the articles cited by the author, should the title of this paper be “Agent-based Social Simulation Require A Boundary”?

---

> ### Author Response · Authors · 2025-11-23
> **Official Response to Reviewer WPiB (1)**
>
> > W1. This paper is purely perspective-based and does not introduce any new techniques or present significant new experiments. It appears to fall outside the scope of the ICLR community and would be better suited for publication in journals or workshops that are more receptive to such work.
>
> We respectfully note that our paper follows the established methodology of position papers, which synthesize existing evidence to establish conceptual frameworks rather than generate new empirical data. Similar to how "The Alignment Problem from a Deep Learning Perspective" (https://openreview.net/forum?id=fh8EYKFKns, ICLR 2024 poster) synthesized extensive literature to establish a framework without conducting original experiments, we systematically analyze documented cases to derive our boundary framework.
>
> > W2. It is questionable whether the three evaluation aspects of alignment, consistency, and robustness proposed in this paper possess any particular characteristics specific to LLM-based social simulation. First, as emphasized by the author throughout the article, consistency and robustness are fundamental requirements for any simulation system. And alignment with the real world remains the ultimate goal of any predictive or simulation effort. Therefore, how does this paper demonstrate the innovation of the proposed evaluation system, and is this evaluation system unique to LLM-based social simulations?
>
> We thank the reviewer for this important critique. We agree with the comment that alignment, consistency, and robustness are fundamental requirements for any simulation system. Nonetheless, we argue that our contribution lies not in inventing these concepts, but in reinterpreting them through the lens of LLM-specific limitations and providing actionable boundaries for LLM-based simulations.
>
> 1. Alignment (Sections 3-4): While alignment is a universal goal, our analysis is specifically tailored to LLM limitations. Regarding traditional ABM alignment, researchers explicitly program behavioral rules and can directly control agent heterogeneity. The challenge is whether the rules capture real mechanisms. Meanwhile, LLM-based simulation alignment faces the "average persona" problem inherent to maximum likelihood training (Section 4.1). Our contribution is to introduce a mean-variance decomposition framework specific to LLM behavior (Section 4.2), and distinguish two critical scenarios (aligned mean/low variance vs. deviated mean/low variance) with different implications and providing guidance on what claims are valid given LLM's limited behavioral variance. This analysis does not apply to traditional ABM because they don't face the "average persona" constraint.
>
> 2. Consistency (Section 5): While temporal consistency matters for all simulations, LLMs face unique challenges. Traditional ABM: State transitions are deterministic or follow predefined probability distributions. LLM agents: Lack genuine memory mechanisms (they rely on context windows), cannot continuously explore environments, and produce responses that may vary inconsistently with slight context perturbations (Yao et al., 2023; Zhu et al., 2023). Our contribution is to identify when this LLM-specific limitation matters (multi-round simulations where behavioral traits significantly affect outcomes) and when it can be safely ignored.
>
> 3. Robustness (Section 6): Again, the challenge manifests differently for LLMs. For Traditional ABM, sensitivity to initial conditions and parameters is well-studied with established methods. While for LLMs, sensitivity to prompt variations, context ordering, and temperature settings is less understood and harder to validate.

---

> ### Author Response · Authors · 2025-11-23
> **Official Response to Reviewer WPiB (2)**
>
> > W3. Throughout this paper, certain sections cite literature from before the advent of LLMs as supporting material, which appears to fall outside the scope of the discussion presented herein.
> >
> > Q3. Based on my understanding of the articles cited by the author, should the title of this paper be “Agent-based Social Simulation Require A Boundary”?
>
> We appreciate the reviewer's careful reading. These two comments are related and highlight an important clarification we need to make.
>
> Our use of pre-LLM literature serves two distinct purposes.
>
> 1. **Social Science foundations.** We need to eastablish Social Science foundations for LLM-based simulations. Sections 2.1 and 3 cite foundational social science literature to establish why heterogeneity and complex interactions matter for social simulations (e.g., Durkheim, 2023; Schelling, 1971; Granovetter, 1978). These principles are timeless and domain-agnostic, defining what makes social simulation meaningful regardless of the technical approach. This literature establishes the evaluation criteria against which we assess whether LLM-based simulations can contribute to social science.
>
> 2. **Comparison between LLM-based simulations and traditional methods.** We need to make a comparison between LLM-based simulations and traditional simulations (which is also related to your W2 and Q2). We cite traditional ABM literature (e.g., Bonabeau, 2002; Epstein, 1999) to show what traditional ABM can and cannot do, how LLM-based approaches differ in their limitations, and why LLM-specific boundaries are needed. For example, in Section 3, we cite Reynolds (1987)'s boids model and Granovetter (1978)'s threshold models to illustrate that simple rules can generate complex collective behavior, which contextualizes why LLM's limited individual variance might still be acceptable for certain phenomena.
>
> **"Outdated methods".** What we do **NOT** do is, we did not apply outdated methods to LLM analysis. All our empirical examples and LLM-specific analysis cite recent literature (2023-2025). Thus, to Q3, we believe "LLM-based" is essential. The boundary problems we identify are LLM-specific. Traditional ABM does not face the "average persona" problem (researchers can explicitly program diverse behaviors), does not suffer from prompt sensitivity or context-window limitations, and allows for interpretable, deterministic state transitions. Our proposed boundaries are tailored to LLM limitations through 3 boundaries. Collective vs. individual boundary is motivated by LLM's low behavioral variance despite alignment attempts, mean-variance framework addresses the "average persona" phenomenon in LLM training, and temporal consistency issue addresses LLM's lack of genuine memory mechanisms.
>
> > Q1. Can an evaluation method be designed based on the proposed evaluation framework to evalulate existing LLM-based social simulation work?
>
> Thank you for this constructive question. Yes, our boundary framework naturally translates into an evaluation approach. We outline the key evaluation steps corresponding to our three boundaries.
>
> **Step 1: Boundary 1 Check - Collective vs. Individual Level**
>
> Question: Does your research focus on collective patterns or individual trajectories?
>
> 1. If collective patterns: Proceed to Step 2.
>
> 2. If individual trajectories: Flag high risk - current LLMs likely unsuitable due to limited behavioral variance.
>
> **Step 2: Boundary 2 Check - Alignment and Heterogeneity**
>
> Question: Do agent behaviors align with real populations? Evaluate using our mean-variance framework (Section 4.2):
>
> 1. Mean alignment test: Compare simulation's average behavior with real-world ground truth.
>     - Statistical comparison (e.g., t-test, effect size analysis).
>     - Expert assessment if quantitative data unavailable.
>
> 2. Variance assessment: Compare behavioral diversity.
>     - Measure distribution spread (e.g., standard deviation, interquartile range).
>     - Check if simulation captures long-tail behaviors when needed.
>
> Classification:
>
> 1. Case 1 (Aligned mean, Low variance): Suitable for qualitative collective pattern analysis.
>
> 2. Case 2 (Deviated mean, Low variance): Simulation unreliable for the target phenomenon
>
> **Step 3: Boundary 3 Check - Consistency and Robustness**
>
> Question: Are simulation results stable and reproducible?
>
> For multi-round simulations (Consistency):
>
> 1. Track agent personality/trait stability across time.
>
> 2. Measure behavioral coherence (stated preferences vs. actions).
>
> 3. If inconsistency detected: limit claims to short-term dynamics.
>
> For all simulations (Robustness):
>
> 1. Sensitivity analysis: vary prompts, parameters (temperature, seeds, etc.).
>
> 2. Check if qualitative patterns remain stable.
>
> 3. If patterns change significantly: simulation fails robustness check.

---

> > ### Author Response · Authors · 2025-11-23
> > **Official Response to Reviewer WPiB (3)**
> >
> > > Q2. Does the evaluation framework proposed in this paper account for the changes introduced by LLM to social simulation?
> >
> > Our framework accounts for changes introduced by LLMs: (1) Section 4.2 discusses how LLM training paradigms create the "average persona" problem (unique to LLMs); (2) Section 5 addresses LLMs' context-dependent inconsistency (not an issue for rule-based agents); (3) Section 6 covers prompt sensitivity (specific to LLM-based systems).
> >
> > We will make these LLM-specific aspects more prominent in our revision.

---

> > > ### Comment · Reviewer_WPiB · 2025-11-26
> > >
> > > Thank you for your response. I will keep my score.

---

> > > > ### Author Response · Authors · 2025-11-27
> > > >
> > > > As we approach the end of the discussion period, please let us know if you have any additional questions or concerns. We appreciate your engagement.

---

> ### Author Response · Authors · 2025-11-26
>
> Thank you for your engagement. Should you have any concerns, please kindly let us know which questions have not been addressed in the above response and why.

---

### Official Review · Reviewer_Z6Uo · 2025-10-18

**Soundness:** 2
**Presentation:** 2
**Contribution:** 1
**Rating:** 2
**Confidence:** 4

**Summary:**

The paper discusses the background of social simulations using computational methods like agent-based modeling (ABM) to understand complex social phenomena, highlighting ABM's limitations in adaptability and representing human-like behaviors. It is motivated by the potential of large language models (LLMs) to create more flexible agents, but argues for establishing boundaries to ensure these simulations reliably contribute to social science by focusing on pattern discovery and hypothesis generation rather than replication or prediction. Key challenges include LLMs' tendency toward an "average persona" with low behavioral heterogeneity, misalignment with real-world patterns, inconsistency over time, and lack of robustness under perturbations. The solutions proposed involve heuristic boundaries emphasizing collective patterns over individual trajectories, alignment with population averages despite limited variance, and rigorous validation methods, along with a practical checklist to guide researchers on appropriate scopes and claims.

**Strengths:**

1. The heuristic boundaries and checklist offer actionable guidance for defining simulation scopes. They focus on collective patterns and validation availability. This bridges AI capabilities with social science needs effectively.

2. The work synthesizes extensive literature to position LLM simulations responsibly. It advocates for avoiding overclaims and focusing on beneficial applications. This fosters interdisciplinary collaboration between AI and social science fields.

**Weaknesses:**

1. Empirical demonstrations are absent as the paper relies solely on critiques of existing studies. No original simulations are conducted to illustrate the proposed boundaries. This limits the ability to verify the framework's practical utility.

2. Potential differences across LLM models are not differentiated in the analysis. Assumptions about universal limitations may not hold for all models or future versions. This could lead to overly broad generalizations.

3. Quantitative metrics for measuring boundaries like variance or alignment are missing. Evaluations rely on qualitative assessments. This hinders objective comparisons across simulations.

4. Accessibility is reduced by dense terminology assuming prior knowledge in both AI and social science. Key concepts like "average persona" could be clarified further. This may limit its reach to interdisciplinary audiences.

5. Overall, the major problem of this work is that its scope is too wide. It is very difficult to figure out the LLM boundary in one work. I think authors should consider shrink the scope, discuss the LLM boundary in some specific aspects and using experiments to provide actionable practical insights, not only the descriptions.

**Questions:**

See Weaknesses.

---

> ### Author Response · Authors · 2025-11-23
> **Official Response to Reviewer Z6Uo (1)**
>
> > W1. Empirical demonstrations are absent as the paper relies solely on critiques of existing studies. No original simulations are conducted to illustrate the proposed boundaries. This limits the ability to verify the framework's practical utility.
>
> **Scope.** We appreciate this suggestion and acknowledge that empirical demonstrations would strengthen the work. Nonetheless, we respectfully note that our paper follows the established methodology of position papers, which synthesize existing evidence to establish conceptual frameworks rather than generate new empirical data. Similar to how "The Alignment Problem from a Deep Learning Perspective" (https://openreview.net/forum?id=fh8EYKFKns, ICLR 2024 poster) synthesized extensive literature to establish a framework without conducting original experiments, we systematically analyze documented cases to derive our boundary framework.
>
> **Empirical evidence.** Further, while we do not conduct new simulations, we ground each boundary in concrete empirical findings from existing work. (1) Alignment (Section 4.2): KBC experiments (Wu et al., 2024) showing aligned peaks but low variance; trolley problem studies (Jin et al., 2024) showing cultural deviation. (2) Consistency (Section 5): Studies of temporal inconsistency (Huang et al., 2024). (3) Robustness (Section 6): Sensitivity studies showing prompt-dependent variations (Zhang et al., 2024a). Each boundary is validated through its ability to explain documented failures across diverse domains.
>
> **Our contribution.** We would also like to emphasize that our contribution lies in organizing scattered empirical findings into a unified framework. Rather than adding one more simulation study, we provide the analytical infrastructure to interpret existing and future studies. That said, we are open to enhancing the paper with a more structured "Framework Application" section systematically analyzing several published studies through our boundary checklist, explicitly showing how it diagnoses their strengths and limitations.
>
> > W2. Potential differences across LLM models are not differentiated in the analysis. Assumptions about universal limitations may not hold for all models or future versions. This could lead to overly broad generalizations.
>
> **Empirical evidence across multiple major model families.** We appreciate this observation and clarify our scope and the evidence supporting our analysis. Our Analysis is grounded in current-generation models with consistent evidence that while we do not exhaustively catalog every model variant (which is not necessary and impossible), our boundary problems are supported by empirical evidence across multiple major model families (please refer to the "empirical findings from existing work" in our response to W1). The consistency of these limitations discussed in this paperacross diverse models suggests they stem from fundamental characteristics of current training paradigms (maximum likelihood estimation on human-generated text, RLHF towards "helpful" averaged responses).
>
> **The generalities of findings.** Our analysis is not merely empirical pattern-matching. We provide theoretical reasons why these boundaries exist. Average persona (Section 4.1) stems from maximum likelihood training objectives that inherently reward high-frequency patterns. This is not model-specific but a consequence of the training paradigm shared across current LLMs. Consistency issues (Section 5) result from LLMs' lack of persistent state and stateless API design. This affects all current LLMs that lack true episodic memory systems. Finally, robustness concerns (Section 6) reflect sensitivity inherent to neural language models' high-dimensional prompt space. These are structural characteristics of current approaches, not accidental features of particular implementations.
>
> **Regarding the claim of "future versions".**  The reviewer suggests our conclusions might not hold for "future versions." However, no position paper can predict future technology, e.g., current LLM limitations don't caveat "but GPT-6 or Gemini 4 might be perfect for this job", and current fairness issues don't caveat "but future models might be unbiased". Our contribution is precisely to identify limitations of current paradigms and provide frameworks to assess future models as they emerge. If future models solve these problems, that validates our framework's utility by showing we correctly identified important challenges.

---

> > ### Author Response · Authors · 2025-11-23
> > **Official Response to Reviewer Z6Uo (2)**
> >
> > > W3. Quantitative metrics for measuring boundaries like variance or alignment are missing. Evaluations rely on qualitative assessments. This hinders objective comparisons across simulations.
> >
> > We acknowledge this issue and explain our rationale for the current approach. On one hand, it is challenging to propose universal quantitative thresholds because social simulation encompasses diverse phenomena (opinion dynamics, market behavior, policy responses, etc.), each requiring **domain-specific metrics**. On the other hand, it is premature to propose universal quantitative thresholds because the field **lacks consensus** on what constitutes "sufficient" alignment or heterogeneity across contexts.
> >
> > We do reference **quantitative evidence** where available, to name a few: Variance comparison in KBC (Figure 2) to measure the degree of alignment, frequency distribution comparison to measure the degree of heterogeneity, correlation coefficients from existing studies to measure the degree of consistency, sensitivity metrics across prompt variations to measure the degree of robustness.
> >
> > We must emphasize that our contribution is the conceptual framework for thinking about boundaries, not prescriptive metrics. The latter require domain-specific development and consensus-building—our framework provides the structure for that process. But we firmly believe that before we can propose universal quantitative thresholds, we need to have a consensus on what constitutes "sufficient" alignment or heterogeneity across contexts.
> >
> > > W4. Accessibility is reduced by dense terminology assuming prior knowledge in both AI and social science. Key concepts like "average persona" could be clarified further. This may limit its reach to interdisciplinary audiences.
> >
> > **Our audience.** We appreciate the concern about accessibility, though we respectfully note some important considerations. LLM-based social simulation is an emerging interdisciplinary research area that inherently requires Machine learning expertise (understanding LLM capabilities, training, evaluation) and Social science methodology (understanding simulation validity, heterogeneity, emergence). Any serious treatment of this topic must engage with both domains. Our paper is positioned for researchers actively working at this intersection—simplifying would undermine rigorous analysis.
> >
> > **Definition of average persona.** Moreover, the reviewer mentions "average persona" as needing clarification. We **explicitly define** this in Section 4.1 which reflects LLMs' built-in bias to
> > converge towards common patterns. It is the tendency of LLMs to generate responses that reflect high-frequency patterns in training data, resulting in agents that behave like "average" humans rather than capturing the full diversity of human behavior. We then develop this concept through theoretical explanation via training objectives, empirical grounding in existing literature, analytical framework (mean-variance decomposition), and concrete implications for simulation validity.
> >
> > > W5. Overall, the major problem of this work is that its scope is too wide. It is very difficult to figure out the LLM boundary in one work. I think authors should consider shrink the scope, discuss the LLM boundary in some specific aspects and using experiments to provide actionable practical insights, not only the descriptions
> >
> > We thank the reviewer for this constructive criticism. We respectfully disagree with the assessment but understand the concern.
> >
> > **Why the boundary problems matter.** The "wideness" of our scope is essential because the three boundary problems (alignment, consistency, robustness) are not independent--they interact in ways that affect overall simulation reliability. For example, low variance (alignment issue) can mask inconsistency problems (consistency issue), and robustness failures may stem from underlying alignment issues. A narrow focus on one aspect would miss these critical interactions.
> >
> > Researchers planning social simulations need to evaluate all these boundaries to determine appropriateness. A framework addressing only one aspect would be incomplete and potentially misleading. Prior work addresses these issues in isolation. Our contribution is precisely the unified framework showing how they collectively define simulation boundaries.
> >
> > **Regarding empirical experiments.** We respectfully argue that comprehensive experiments across all boundaries would be infeasible and potentially less valuable. (1) Properly validating alignment, consistency, and robustness across diverse social phenomena (economics, sociology, psychology, policy) would require dozens of studies--beyond the scope of any single paper. (2) Such experiments would largely replicate existing published studies, which we systematically synthesize. (3) Our contribution is analytical synthesis rather than empirical demonstration. We provide the framework to interpret existing and future empirical work.

---

> ### Author Response · Authors · 2025-11-27
>
> As we approach the end of the discussion period, we kindly ask whether the response has addressed your concerns. Your engagement will be appreciated.

---

### Official Review · Reviewer_5MeJ · 2025-10-30

**Soundness:** 1
**Presentation:** 2
**Contribution:** 1
**Rating:** 2
**Confidence:** 4

**Summary:**

This paper presents a survey and reflection on large language model-based social simulation. The authors review existing studies, discuss the boundaries of what large language model-based social simulations can or cannot achieve, and propose a checklist of methodological considerations such as alignment, temporal consistency, and robustness. The stated goal of the paper is to help researchers design more credible and interpretable social simulations that can contribute to social scientific understanding rather than merely reproducing known behaviors.

**Strengths:**

* The topic is timely. Large language model-based social simulation is a rapidly emerging area that indeed requires systematic methodological reflection.

* The paper is clearly written and well-organized.

* The authors try to connect social scientific perspectives with large language model agent research, which could be informative for newcomers to the field.

**Weaknesses:**

1. Lack of novel research contribution.

The paper primarily summarizes and comments on existing works. It does not introduce a new theoretical framework, formal model, or empirical study. ICLR normally expects some form of novel insight, methodology, or evaluation rather than a descriptive review.

2. Insufficient depth and evaluation.

The proposed “boundary checklist” remains conceptual and is not validated through adequate case studies or quantitative analysis. Without such evidence, it is difficult to assess its utility or correctness.

3. Ambiguous positioning relative to ICLR scope.

The work fits better as a perspective or survey article rather than a research contribution in machine learning. ICLR generally values algorithmic, theoretical, or experimental advances, and this paper focuses more on methodological reflection and normative guidance.

4. Some important literature is missing.

The paper omits several important studies. For example,

[1] Gabriel, Iason, et al. "Who’s to blame when AI agents mess up? We urgently need a new system of ethics." Nature (2025): 38-40.

[2] Kozlowski, Austin C., and James Evans. "Simulating Subjects: The Promise and Peril of Artificial Intelligence Stand-Ins for Social Agents and Interactions." Sociological Methods & Research (2025): 00491241251337316.

[3] Grossmann, Igor, et al. "AI and the transformation of social science research." Science 380.6650 (2023): 1108-1109.

[4] Bail, Christopher A. "Can Generative AI improve social science?." Proceedings of the National Academy of Sciences 121.21 (2024): e2314021121.

**Questions:**

As the field of large language model-based agents for social science is still emerging, there are already several pioneering studies that reflect on its implications for social research and highlight methodological caveats when using such simulations. Some of these important works are missing from the current paper (for example, those listed above). Could the authors clarify how their contribution differs from and advances beyond these prior reflections, both conceptually and methodologically?

---

> ### Author Response · Authors · 2025-11-23
> **Official Response to Reviewer 5MeJ (1)**
>
> > W1. Lack of novel research contribution. The paper primarily summarizes and comments on existing works. It does not introduce a new theoretical framework, formal model, or empirical study. ICLR normally expects some form of novel insight, methodology, or evaluation rather than a descriptive review.
>
> We respectfully disagree with the assessment that position papers lack novelty or are unsuitable for ICLR. Position papers that provide critical reflection and normative guidance have been accepted at ICLR. For example, "The Alignment Problem from a Deep Learning Perspective" (https://openreview.net/forum?id=fh8EYKFKns, ICLR 2024 poster) similarly provides conceptual frameworks and checklists without introducing new algorithms or empirical evaluations. Like our work, it offers normative guidance on when and how to responsibly apply ML techniques.
>
> **Our contribution.** While we build on existing critiques of LLM-based simulations, we provide novel contributions:
>
> 1. A unified analytical framework: distinguishing three boundary problems (alignment, consistency, robustness) that determine simulation reliability for social pattern discovery, which differs from prior descriptive taxonomies of problems.
>
> 2. A mean-variance decomposition framework (Section 4) for diagnosing different types of alignment failures, with specific implications for each scenario.
>
> 3. Theoretical grounding for why heterogeneity is fundamental to social simulation, drawing from complex systems theory and connecting it to LLMs' "average persona" problem (Section 3).
>
> 4. Actionable heuristic boundaries (Section 7) that provide researchers with concrete criteria for determining when LLM simulations can make reliable contributions.
>
> The contribution is not a new algorithm but a conceptual framework that advances how the field should think about and apply LLM-based simulations, precisely what position papers are designed to provide.
>
> > W2. Insufficient depth and evaluation. The proposed “boundary checklist” remains conceptual and is not validated through adequate case studies or quantitative analysis. Without such evidence, it is difficult to assess its utility or correctness.
>
> We appreciate this concern and acknowledge that empirical validation would strengthen the work. Nonetheless, we note several points:
>
> 1. **The purpose of this work.** Position papers establish conceptual frameworks and normative guidelines. The "boundary checklist" is intentionally conceptual because it provides heuristic guidance applicable across diverse social phenomena. Rigid quantitative thresholds would be inappropriate given the variety of research contexts. The goal is to promote critical reflection, not to provide algorithmic decision rules.
>
> 2. **Empirical evidence.** While we do not present new case studies, we extensively cite and analyze concrete examples from existing literature: Economic simulations (Section 4.2, Case 1): KBC experiments showing aligned peaks but low variance (Wu et al., 2024); Cultural bias issues (Section 4.2, Case 2): Trolley problem studies (Jin et al., 2024); Consistency problems (Section 5): Studies documenting temporal inconsistency (Huang et al., 2024) These examples substantiate each boundary problem we identify.
>
> 3. **Our contribution.** The validation of our framework comes from its ability to explain and organize existing empirical findings. Each boundary problem we identify is grounded in documented failures or limitations in prior work. Our contribution is synthesizing these scattered findings into a coherent framework.
>
> We are happy to add a more structured case study analysis in the next version, systematically applying our checklist to 3-5 published LLM simulation studies to demonstrate its diagnostic value. This would show how our framework helps identify which studies make appropriate vs. overclaimed contributions, where validation was insufficient, and what types of phenomena were suitable vs. unsuitable for the simulation approach used.

---

> > ### Author Response · Authors · 2025-11-23
> > **Official Response to Reviewer 5MeJ (2)**
> >
> > > W3. Ambiguous positioning relative to ICLR scope. The work fits better as a perspective or survey article rather than a research contribution in machine learning. ICLR generally values algorithmic, theoretical, or experimental advances, and this paper focuses more on methodological reflection and normative guidance.
> >
> > For the relevance of position papers to ICLR scope, please refer to our response to W1.
> >
> > In addition, we believe this work is well-suited for ICLR for several reasons, (1) Recent ICLR conferences have increasingly featured work on LLM agents, multi-agent systems, and AI for social applications. Our paper directly addresses the reliability and validity of a rapidly growing subfield within LLM research. (2) Just as papers on fairness, interpretability, and robustness provide essential methodological guidance for ML applications, our work establishes methodological foundations for applying LLMs to social simulation, an area where inappropriate use could lead to misleading scientific conclusions. (3) Our analysis of boundary problems (especially the mean-variance framework and heterogeneity requirements) points to specific technical challenges that require algorithmic innovation, e.g., methods to enhance behavioral variance in LLM outputs, techniques for maintaining consistency in long-horizon simulations, and approaches for validating alignment with real human distributions. This directly informs the ML research agenda.
> >
> > > W4. Some important literature is missing.[...]
> > >
> > > Q1. Could the authors clarify how their contribution differs from and advances beyond these prior reflections, both conceptually and methodologically?
> >
> > We thank the reviewer for highlighting these important works. We have carefully reviewed all suggested papers and deeply respect these contributions. We will incorporate appropriate references in the next version. Nonetheless, we must clarify that our contribution is **NOT "another review of LLM limitations"** but rather focuses on the following aspects compared to the existing literature:
> >
> > - A theoretical account of why heterogeneity is central to social simulation.
> >
> > - An analytical framework (mean-variance decomposition) for diagnosing validity.
> >
> > - A normative checklist defining boundary conditions for valid claims.
> >
> > The existing literature provides valuable foundations while focusing on different aspects than our work.
> >
> > 1. Kozlowski & Evans (2025): this comprehensive methodological guide addresses how to simulate human subjects with LLMs, covering design principles, six limitations, and validation strategies. While they offer a toolkit for conducting simulations, we provide principles for determining when the toolkit should be applied.
> >
> > 2. Bail (2024): this PNAS perspective provides a broad survey of LLM applications across social science, including opportunities (survey research, experiments, text analysis, agent-based models), limitations (bias, junk science, ethical concerns, replication challenges), and solutions (building open-source infrastructure for social science). It reviews catalogues what LLMs can do and what can go wrong. Meanwhile, we provide analytical tools to determine the specific validity boundaries, e.g., theoretical foundation for why certain limitations (especially heterogeneity) are fundamental, diagnostic framework to assess when simulations produce valid vs. invalid inferences, boundary conditions defining the scope of defensible claims.
> >
> > 3. Grossmann et al. (2023): this Science perspective offers a high-level overview of AI's transformative potential, including applications (LLMs as surrogates, confederates, and research aids), scientist-humanist dilemma (tension between studying unbiased models vs. engineering ethical AI), and trade-offs (external vs. internal validity, ethical considerations). Meanwhile, we provide systematic frameworks rather than high-level observations, e.g., theoretical foundation for why heterogeneity matters, quantitative diagnostics to answer validity questions, and concrete guidance on research design choices.
> >
> > 4. Gabriel et al. (2025): this Nature comment addresses ethical challenges of deploying AI agents in the real world, including alignment problems (agents misinterpreting instructions, taking shortcuts), social relationships (human-AI companionship, emotional manipulation risks), and governance needs (guard rails, authorization protocols, multi-agent ecosystem regulation). The deployment focuses on real-world interactions, while we focus on research applications where AI simulates human subjects for scientific inquiry.
> >
> > Our focus is on **when and under what conditions** simulations support valid scientific claims. This is a distinct contribution that complements rather than duplicates existing work. We will revise our introduction to better position our work relative to these important antecedents, emphasizing how our boundary-setting framework complements existing methodological guidance and application surveys.

---

> ### Author Response · Authors · 2025-11-27
>
> As we approach the end of the discussion period, we kindly ask whether the response has addressed your concerns. Your engagement will be appreciated.

---

### Official Review · Reviewer_WDSf · 2025-10-31

**Soundness:** 3
**Presentation:** 2
**Contribution:** 2
**Rating:** 4
**Confidence:** 3

**Summary:**

This paper presents a position and analysis of the use of large language models (LLMs) for social simulation. The authors argue that for LLM-based simulations to be meaningful to social science, researchers must recognize and operate within clear boundaries. The paper identifies three primary "boundary problems": alignment (do simulated patterns match reality?), consistency (does an agent maintain its persona/behavior over time?), and robustness (is the simulation reproducible?).

The paper's core thesis is that current LLMs are constrained by an "average persona" that exhibits low behavioral variance (lacks heterogeneity), which is a fundamental limitation for simulating complex social dynamics. The authors use a mean-variance framework to analyze this alignment problem. They conclude by proposing "heuristic boundaries," chiefly that researchers should focus on collective patterns rather than individual agent trajectories, and that claims are most reliable when the mean of the collective behavior aligns with human data, even if the variance is low.

**Strengths:**

1. The paper's greatest strength is its clear, concise, and persuasive writing. It's an excellent summary of the key challenges in the field.

2. The "mean-variance" analysis of the alignment problem is a useful and intuitive way to decompose the "average persona" issue.

3. The paper's central message—that the field must be more critical, define its boundaries, and move beyond simple "replication"—is a crucial and timely corrective for the community.

4. The final "heuristic boundaries" (e.g., "focus on collective patterns, not individual trajectories") are practical, actionable, and well-justified by the preceding analysis.

**Weaknesses:**

1. The primary weakness is the paper's failure to acknowledge and differentiate itself from
What Limits LLM-based Human Simulation: LLMs or Our Design? arXiv:2501.08579. This prior work identified the exact same "LLM-inherent limitations" (lack of diversity/heterogeneity and inconsistency) as the core bottlenecks. This submission does not offer a new conceptual leap beyond what is already present in that paper. For a position paper, where the idea is the main contribution, this overlap is a critical flaw. And this paper does not cite that paper.

2. As a position paper, it excels at identifying problems (lack of heterogeneity, consistency, robustness) but offers limited, high-level solutions. It does not propose new methods for how to measure robustness or how to quantitatively establish an "aligned mean," which are left as open challenges.

**Questions:**

1. Can you please explicitly state what the novel intellectual contribution of this paper is beyond the critiques already raised in that prior work? Why is your "boundary problem" framework (alignment, consistency, robustness) a more generative or insightful model than the "LLM-inherent vs. Design" framework?

2. You identify the "average persona" (low variance) as a core issue. Do you believe this is a fundamental and perhaps unsolvable limitation of the current maximum likelihood training paradigm? Or do you see it as a solvable engineering problem that can be addressed with better sampling, prompting, or fine-tuning techniques?


3. You propose focusing on "collective patterns." However, many crucial social phenomena (e.g., innovation diffusion, radicalization, market panics) are specifically driven by atypical agents or "long-tail" behaviors, not the average. Does your analysis imply that this entire class of social phenomena is fundamentally outside the boundary of what LLMs can reliably simulate today?

---

> ### Author Response · Authors · 2025-11-23
> **Official Response to Reviewer WDSf (1)**
>
> We appreciate the reviewers' recognition of the merits of our work. Our responses to the reviewers' concerns and questions are as follows:
>
> > W1. The primary weakness is the paper's failure to acknowledge and differentiate itself from What Limits LLM-based Human Simulation: LLMs or Our Design? arXiv:2501.08579. This prior work identified the exact same "LLM-inherent limitations" (lack of diversity/heterogeneity and inconsistency) as the core bottlenecks. This submission does not offer a new conceptual leap beyond what is already present in that paper. For a position paper, where the idea is the main contribution, this overlap is a critical flaw. And this paper does not cite that paper.
> >
> > Q1. Can you please explicitly state what the novel intellectual contribution of this paper is beyond the critiques already raised in that prior work? Why is your "boundary problem" framework (alignment, consistency, robustness) a more generative or insightful model than the "LLM-inherent vs. Design" framework?
>
> We would like to clarify that we did cite the paper in our submission (please refer to lines 146 and 860, as Wang et al., 2025c). In addition, we respectfully argue that our paper makes fundamentally different contributions:
>
> 1. Wang et al. provided a comprehensive survey of LLM-based human simulation across four domains, asking whether it is LLMs or the design that limits the simulation. This is a descriptive taxonomy classifying problems into "LLM-inherent" vs. "design" issues. While our paper focuses specifically on establishing **when and how** LLM-based simulations can meaningfully **contribute to social science research** by defining applicable boundaries, and we propose a normative framework defining three critical boundaries (alignment, consistency, robustness) that determine the **reliability of social pattern discovery**, which is the ultimate goal of social simulation.
>
> 2. While both papers identify lack of heterogeneity as a limitation, we systematically argue from the perspective of complex systems **why heterogeneity is fundamental** for social simulation (Section 3), introduce a **mean-variance decomposition framework** to diagnose different types of alignment problems (Section 4), and distinguish between two critical scenarios with different implications for simulation validity.
>
> Thus, we believe Wang et al. and our paper are complementary: Wang et al. comprehensively analyzed the cause of the limitation of LLM-based simulations, while we provide normative guidance on when and how to responsibly apply LLM simulations in social science research.

---

> > ### Author Response · Authors · 2025-11-23
> > **Official Response to Reviewer WDSf (2)**
> >
> > > W2. As a position paper, it excels at identifying problems (lack of heterogeneity, consistency, robustness) but offers limited, high-level solutions. It does not propose new methods for how to measure robustness or how to quantitatively establish an "aligned mean," which are left as open challenges.
> > >
> > > Q2. You identify the "average persona" (low variance) as a core issue. Do you believe this is a fundamental and perhaps unsolvable limitation of the current maximum likelihood training paradigm? Or do you see it as a solvable engineering problem that can be addressed with better sampling, prompting, or fine-tuning techniques?
> >
> > Thank you for this insightful question. We view the "average persona" problem as **partially mitigatable but fundamentally challenging within current paradigms**.
> >
> > We would like to explain why we claim the solution in the way it was presented in our paper.
> >
> > 1. As discussed in Section 4.1, the maximum likelihood training objective inherently rewards high-frequency patterns and suppresses marginal ones. This creates an inherent tension that the training process, while making LLMs useful for general purposes, at the same time limits their distributional representativeness.
> >
> > 2. We argue that current mitigation strategies are insufficient. Section 4.2 reviews existing approaches:
> >     - Prompt engineering cannot completely eliminate bias, especially for minority groups.
> >     - Personality-based methods often capture only superficial stereotypes.
> >     - Data-driven alignment requires large-scale personalized data that is costly and scarce.
> >     - Multi-model ensembles still show limited heterogeneity.
> >
> > 3. Thus, rather than claiming this is "unsolvable," we argue that researchers should **accept** this limitation and work within it by focusing on phenomena where aligned mean is sufficient in spite of low variance (Case 1 in Section 4.2), and clearly **identify** when phenomena require heterogeneity that current LLMs cannot provide (Case 2). Meanwhile, the researchers should **avoid overclaiming**,  by focusing on collective patterns rather than individual trajectories (Boundary 1, Section 7).
> >
> > We believe future advances (e.g., training with more diverse data, Section 5.2; better personalization techniques, Section 8) may gradually improve heterogeneity, but this remains an active research challenge rather than a solved problem. Thus, the key contribution of our paper is not to solve this problem but to delineate clear boundaries so researchers can make responsible claims about what current LLMs can and cannot reliably simulate.

---

> > > ### Author Response · Authors · 2025-11-23
> > > **Official Response to Reviewer WDSf (3)**
> > >
> > > > Q3. You propose focusing on "collective patterns." However, many crucial social phenomena (e.g., innovation diffusion, radicalization, market panics) are specifically driven by atypical agents or "long-tail" behaviors, not the average. Does your analysis imply that this entire class of social phenomena is fundamentally outside the boundary of what LLMs can reliably simulate today?
> > >
> > > We thank the reviewer for the insightful and challenging question that highlights an important limitation of current LLM-based simulations.
> > >
> > > Our answer to your question is: yes, we do believe that phenomena fundamentally driven by long-tail behaviors are largely outside the reliable boundary of current LLM simulations. Nonetheless, we want to clarify several nuances.
> > >
> > > 1. We need to distinguish the purpose of the simulation. Our argument focuses on simulations aimed at **discovering and validating general social patterns**. For phenomena like innovation diffusion or radicalization: (1) if the research goal is to understand how a small number of innovators/radicals trigger widespread adoption through specific mechanisms (e.g., threshold models, network effects), and the focus is on the emergent collective dynamics rather than precise individual trajectories, then LLM simulations might still offer value if the mean behavior aligns reasonably well (but based on the current LLM, it is not guaranteed to be reliable). (2) If the research goal requires accurately simulating the specific characteristics and decision-making processes of atypical agents (e.g., "who becomes a radical?", "what drives contrarian traders?"), then yes, current LLMs are inadequate due to their tendency toward average personas.
> > >
> > > 2. This is a critical boundary that researchers must carefully examine whether their phenomena of interest fall within the boundary. Phenomena requiring authentic long-tail behavior represent an important class of social dynamics that currently exceed the boundary of reliable LLM-based simulations. This is precisely why we advocate for (1) clear boundary delineation (Section 7), (2) rigorous validation against real-world heterogeneity (Section 6), and (3) transparent acknowledgment of simulation limitations.
> > >
> > > 3. We acknowledge that this is an open challenge and we will continue to research on this topic. Until substantial progress is made, researchers should avoid using LLM simulations for phenomena fundamentally requiring long-tail behaviors, and alternative methods (traditional ABM with carefully designed heterogeneity, or hybrid approaches) may be more appropriate.
> > >
> > > Thank you for helping us sharpen this critical point.

---

> > > > ### Comment · Reviewer_WDSf · 2025-11-25
> > > >
> > > > Your rebuttal addresses my concerns. I increased to score to 6.

---

> > > > > ### Author Response · Authors · 2025-11-25
> > > > > **Response to Reviewer's Engagement**
> > > > >
> > > > > Thank you for taking the time to review our rebuttal. We truly appreciate your helpful comments and suggestions. We will include a discussion of the points you raised, especially the takeaway on the average-persona issue and long-tail behaviors in the next version.

---

### Author Response · Authors · 2025-12-02
**Author Remarks**

Dear Area Chair,

We sincerely appreciate your time and effort in meta-reviewing our paper.

**TL;DR** This work establishes a normative framework for LLM-based social simulations, defining critical boundaries regarding alignment, consistency, and robustness. We provide actionable guidelines to address LLMs' inherent lack of behavioral diversity, ensuring they are reliably used for social pattern discovery rather than uncritical replication.

**Summary of Contributions.** This work establishes a normative framework to determine when and how LLM-based simulations can reliably contribute to social science. Unlike existing descriptive surveys, we provide:

1. **Three Heuristic Boundaries:** A prescriptive checklist covering Alignment, Consistency, and Robustness to prevent scientific overclaims.

2. **Mean-Variance Decomposition Framework:** A novel theoretical lens (Section 4) to diagnose the "average persona" problem, distinguishing between acceptable low-variance scenarios (Case 1) and invalid deviated-mean scenarios (Case 2).

3. **Theoretical Grounding:** We synthesize complex systems theory with LLM training limitations (e.g., Maximum Likelihood Estimation) to explain why heterogeneity is the fundamental bottleneck for simulation validity.

---

**Highlights Recognized by Reviewers**

1. **Impact:** The proposed boundaries of LLM-based simulations are crucial and timely, providing guidance for relevant researchers (Reviewers WDSf, 5MeJ, and Z6Uo).

2. **Audience:** The paper connects bridges AI capabilities with social science needs, informative for newcomers to the field (Reviewer 5MeJ and Z6Uo).

3. **Presentation:** The paper is clearly written and well-organized (Reviewers WDSf, 5MeJ, and WPiB).

4. **References:** The paper synthesizes extensive literature (Reviewer Z6Uo).

---

**Discussion Outcome**

Reviewer WDSf has acknowledged that we **successfully addressed the concerns** regarding the distinction from prior work (Wang et al., 2025c) and has subsequently **raised the score to 6** on Nov. 24, 19:51, UTC-12 (AoE), prior to OpenReview's system issue widely spread over the internet.

Next, we summarize our responses to reviewers' major concerns regarding the scope and nature of this position paper.

**Scope and Novelty:** Reviewers 5MeJ (W1 & W3) and WPiB (W1) commented that our paper lacks contribution because it does not propose a new algorithm or dataset.

[Response] We strongly respectfully disagree with this assessment, because our paper is not "another review of LLM limitations".

1. *Validity of Position Papers:* Position papers with conceptual frameworks are a standard and vital category of contribution at ICLR (e.g., The Alignment Problem from a Deep Learning Perspective, https://openreview.net/forum?id=fh8EYKFKns, accepted to ICLR 2024). As LLM agents are increasingly used for scientific inquiry, methodological guardrails are an urgent prerequisite for validity.

2. *Distinct Intellectual Contribution:* Our contribution is not a literature review, but the analytical infrastructure (specifically the mean-variance framework and the heterogeneity-based validity criteria) that allows researchers to interpret empirical results.

**Empirical Demonstrations:** Reviewers 5MeJ (W2) and Z6Uo (W1) commented that empirical demonstrations are absent as the paper relies solely on critiques of existing studies.

[Response] We clarified that our methodology of synthesizing extensive existing empirical evidence (e.g., Wu et al., 2024; Jin et al., 2024; Huang et al., 2024) to derive a unified framework is appropriate for a position paper. Validating our framework does not require running a new, singular simulation, but rather demonstrating its ability to explain the failure modes and successes across the breadth of the existing literature, which we have done.

**Differentiation from Prior Works:** Reviewers WDSf (W1 & Q1) and 5MeJ (W4 & Q1) commented that some important literature is missing.

[Response] While prior works (e.g., Wang et al., 2025c) provide a taxonomy of limitations, our work provides normative boundaries defining the conditions under which valid scientific claims can be made. In addition, while we appreciate the literature suggestions by Reviewer 5MeJ, the cited references do not directly align with the focus of our study.

---

Again, we appreciate the area chair and the reviewers for their efforts in reviewing our work. We believe this work offers a timely and necessary intervention to ensure the rigor of the growing field of LLM-based computational social science.


Best wishes,

Authors of Submission 11711

---

### Meta-Review · Area_Chair_hGsQ · 2026-01-08

**Summary:**

This paper presents a position paper arguing that LLM-based social simulations must operate within clearly defined boundaries. Reviewers broadly agree that the topic is timely, the paper is clearly written, and the proposed checklist offers intuitive, high-level guidance. However, the reviewers also raise the concerns that the contribution is primarily reflective rather than novel, with substantial overlap with prior position papers. Multiple reviewers also questioned the breadth of scope and the absence of concrete metrics or demonstrations. Given these concerns, I recommend rejection.

**Reviewer Concerns:**

The rebuttal partially addresses concerns about positioning and differentiation.

**Reviewer Scores:**

Reviewer WDSf will raise score to 6.

---

### Decision · Program_Chairs · 2026-01-26

Reject